# Advancements in Crop PUFAs Biosynthesis and Genetic Engineering: A Systematic and Mixed Review System

**DOI:** 10.3390/ijms26083462

**Published:** 2025-04-08

**Authors:** Molalign Assefa, Yajie Zhao, Chao Zhou, Yuanda Song, Xiangyu Zhao

**Affiliations:** 1Colin Ratledge Center for Microbial Lipids, School of Agricultural Engineering and Food Science, Shandong University of Technology, Zibo 255000, China; molalignassefa@sdut.edu.cn; 2State Key Laboratory of Crop Biology, College of Life Sciences, Shandong Agricultural University, Taian 271018, China; 198591yy@163.com (Y.Z.); zhouc@sdau.edu.cn (C.Z.)

**Keywords:** crop biotechnology, essential lipids, healthcare, nutrition, PUFA

## Abstract

Recent advances in molecular studies on plant lipids have revealed novel functions, increasing interest in their roles in plant metabolic processes and food functionality. With evolving living standards, the demand for crop-derived polyunsaturated fatty acids (PUFAs) oil is increasing due to their benefits for cardiovascular health, brain function, and anti-inflammatory properties. Despite these benefits, there are gaps in comprehensive, integrated, and consolidated documents on recent advancements in crop biotechlogy, particularly concerning the biosynthesis of essential lipids. Such a document could provide valuable insights for researchers, breeders, and industry professionals seeking to enhance crop oil profiles and optimize the nutritional and functional qualities of plant-based foods. Therefore, this study aims to: (1) provide an updated review of crop lipid biosynthesis and (2) identify trending topics, key contributors, and institutions contributing to research on crop PUFAs, their health benefits, and genes associated with these functions. Methods: Systematic and mixed-method review approaches were used to gather the most recent evidence by identifying all relevant primary research studies on the specific review topic. Five databases were used in the process. Result and conclusion: 366 papers were identified, with 73 highly cited and recent ones focusing on crop PUFA biosynthesis and genetic engineering. Key genes involved in lipid biosynthesis include *FAD*, *TMT*, *HGG*, *GhKAR*, *GhHAD*, and transcription factors like *MYB89*, *MYB96*, *WRI*, *LEC*, *GL2*, *FUS3,* and *HB2* all critical for enhancing PUFA biosynthesis. However, challenges such as poor transgene expression, reduced seed germination, and metabolic toxicity must be addressed to develop crops with improved oil profiles.

## 1. Introduction

Researchers across the health and agricultural sectors are increasingly emphasizing the importance of evidence-based practices, both in healthcare and crop molecular research. This growing focus on evidence-based approaches is particularly evident in the fields of food science and nutrition, where nutrition-related practices are guided by evidence-based frameworks and resources, which are closely related to ongoing research [1,2]. In healthcare, evidence-based practices are driving the shift toward more personalized, preventative, and efficient treatments, with a clear focus on improving patient outcomes through scientifically validated methods [3]. Similarly, in crop molecular research, evidence-based strategies are being applied to enhance the nutritional quality and functionality of crops, optimizing plant breeding techniques, and improving the health benefits of agricultural products [4,5]. The convergence of these two fields—healthcare and biotechnology—reflects a broader recognition of the critical role that both plant-based foods and nutrition play in human health. Researchers are now working to bridge the gap between food production and health outcomes, creating a more holistic approach to managing public health through improved dietary practices and functional foods [6]. A comprehensive understanding of the nutritional and biological functions of lipids has remained a central focus of international research efforts (Figure 1). The rising incidence of lifestyle-related diseases, such as cardiovascular diseases, hypertension, diabetes, and other conditions closely linked to the quantity and quality of dietary fat consumption, presents a growing concern. Regulating diets to achieve an optimal balance between saturated and polyunsaturated fats offers significant health benefits [2].

Over the past several decades, significant advances in genomics and molecular biology have led to the identification of key genes involved in the synthesis of fatty acids (FAs) in plants [7,8,9,10,11,12,13,14]. These breakthroughs have provided a deeper understanding of the complex biochemical pathways that govern lipid metabolism, including the enzymes responsible for the elongation, desaturation, and modification of fatty acids [15]. Armed with this knowledge, researchers have been able to develop transgenic oil crops with enhanced or tailored fatty acid profiles, allowing for the creation of plants that produce oils with specific functional properties. These genetically engineered crops can be designed to yield oils with improved nutritional quality, such as higher levels of polyunsaturated fatty acids (PUFAs) or monounsaturated fatty acids (MUFAs), which are beneficial for human health [16]. Additionally, transgenic crops can be engineered to produce oils with altered fatty acid compositions that are more suitable for industrial applications, such as biofuels, or for use in processed foods with longer shelf lives. The ability to manipulate fatty acid biosynthesis through genetic engineering has thus opened new avenues for improving crop performance, enhancing food quality, and meeting the demands of both the food and biofuel industries.

Utilizing tools such as molecular techniques, hybrid production technology, and double haploids [7], scientists have made significant advancements. These genetically engineered oil crops hold great promise for creating new and improved varieties with desirable traits, including increased oil content and modified oil profiles. Such crops offer unique qualities, not only for food production but also for oleochemicals, feedstock, and biofuels. Additionally, they present economically feasible sources of enhanced biofuels, therapeutic compounds, and varieties tolerant to abiotic and biotic stresses as well [17].

Fatty acids are synthesized de novo in the plastid, using *acetyl-CoA* derived from photosynthesis as precursor and *malonyl-ACP* as elongator [18]. Malonyl-thioester undergoes a series of condensation reactions with an *acetyl-CoA* catalyzed by a *3-ketoacyl-ACP synthase-III (KCSIII)* that produces propionyl-ACP (C4:0-ACP). Subsequent condensation reaction takes place up to the formation of palmitoyl-ACP (C16:0-ACP) catalyzed by a *KAS-I* isomer. Finally, *KAS-II* elongates the C16:0-ACP to stearoyl-ACP (C18:0-ACP), then *FAD6,* a desaturase enzyme, catalyzes C18:0-ACP to C18:1-ACP, and *FAD37/FAD8* follows to catalyze the linoleic acid to linolenic acid [14]. The three KAS components are *KASI*, the initial reaction between Acyl-CoA and *malonyl-ACP*; *KASII*, the sequential elongation till C16:0-ACP; and *KASIII* or *FAB1* for the addition of an extension to C18:-ACP.

After fatty acid synthesis in the plastid, the fatty acyl-ACP moieties, mainly palmitoyl, stearoyl, and oleoyl-ACPs, are either used directly for lipid biosynthesis in the plastid or can be hydrolyzed by *fatty acyl-ACP thioesterases (FATA/FATTB)* to free fatty acid. They are later exposed to the endoplasmic reticulum in the form of the acyl-CoA pool. The acyl pool undergoes several modifications regarding elongation, desaturation, and exchange catalyzed by different endoplasmic reticulum membrane-bound proteins, which constitute the endoplasmic reticulum pathways of lipid biosynthesis. It comprises four major enzymes: *KCS*, *KCR*, *HCD,* and *ECR*, all of which function in all tissues exhibiting VLFA biosynthesis and possess broad substrate specificity. Therefore, following the hydrolysis of *acyl-ACP* by plastid membrane-localized *FATA/B thioesterase*, 16 or 18 carbon fatty *acyl-CoAs* are produced. These fatty *acyl-CoAs* are then transported to the endoplasmic reticulum, where they can be elongated by a microsomal *fatty acid elongation (FAE*) complex/metabolon, comprised of the four enzymes catalyzing four successive reactions [19]. The reactions are (1) the condensation of *malonyl-CoA* with an *acyl-CoA* catalyzed by a *3 ketoacyl-CoA synthase (KCS)*, *to provide an output* β-oxoacyl-CoA; (2) the reduction of β *-ketoacyl-CoA* by a *3-ketoacyl-CoA reductase (KCR)*, to attain β-hydroxyacyl-CoA; (3) the dehydration of β- *hydroxyacyl-CoA* by a *3-hydroxyacyl-CoA dehydratase* (*HCD)* to attain enoyl-CoA; and (4) the reduction of *enoyl-CoA* by an *enoyl-CoA reductase* (*ECR*) (Figure 2), resulting in acyl-*CoA* with two more carbons in the chain, which can be further elongated up to 32 carbons [20]. To sum up, linoleic acid (LA: 18:2n-6 or 18:2Δ9, 12) and alpha-linolenic acid (ALA: 18:3∆9, 12, 15) serve as precursors for the synthesis of VLCPUFAs (see Figure 2), which are fatty acids with 20 or more carbons and double bonds in the cis configuration [21]. Following this, acyl-CoAs are sequentially transferred to glycerol-3-phosphate (G3P) by *acyltransferase* enzyme in the ER, resulting in the formation of triacylglycerol (TAG).

Although debated, the expression and regulation of the mentioned fatty acid-related genes are influenced by interactions with transcription factors (TFs) [22]. The research conducted by Huang and his team found that RNA-seq and RT-qPCR analyses indicated differential expression of *pfbZIP*, basic leucine zipper genes, during the seed development of *Perilla frutescens*, a novel oilseed crop. Additionally, several *WRINKLED1* (*Wheat*), *pfbZIP* (*P. frutescens*) genes showed significant correlations with key oil-related genes [23].

Significant attention is given to PUFAs, particularly omega-3 (ω-3) and omega-6 (ω-6) fatty acids, due to their multi-health and industrial benefits. Recent research has focused on enhancing the biosynthesis of ω-3 and ω-6 fatty acids through molecular engineering of crops such as chia, perila, soybean, sunflower, flax, and other vegetable oil sources. Among ω-3 fatty acids, ALA, EPA, and DHA play essential roles in supporting brain function and overall health [24]. In general, recent advances in molecular research on plant lipids have revealed novel functions, with growing interest among plant scientists in understanding how lipids impact plant functions and contribute to food functionality. Research on plant seed oils has gained attention due to population growth, rising living standards, and the need for antioxidants that neutralize harmful free radicals and reactive oxygen species [25].Therefore, a detailed mixed-method approach with a systematic review is needed to gain a comprehensive understanding of published research and citation trends in the field of crop lipid biosynthesis. This approach will also provide valuable insights into current progress in crop research focusing on lipid biosynthesis pathways. Therefore, the current study has the following main objectives: (1) to provide a consolidated update from leading experts, serving as the foundation for future research and guidance for new researchers; and (2) to identify trending topics, key authors, and institutions contributing to research on crop PUFAs, their health benefits, and genes associated with these functions as identified by scholars.

## 2. Materials and Methods

### 2.1. Systematic and Mixed with Bibliometric Reviews: Key Methods in Evidence Synthesis

Systematic and other review methodologies like bibliometrics are essential tools for synthesizing the most current and comprehensive evidence on a given topic. By systematically identifying all primary research studies relevant to a specific area of investigation, these methods ensure that the latest findings are included in the review process [26]. Bibliometric approaches, in particular, allow for the quantitative analysis of research trends, author contributions, citation patterns, and institutional networks, providing valuable insights into the scope and impact of research in a field [27]. Alongside this, traditional review methodologies critically evaluate the quality and reliability of the selected studies, assessing the methodologies, results, and conclusions drawn in each piece of research. These methods aim to not only summarize existing evidence but also to integrate and synthesize findings from diverse studies, offering a more cohesive understanding of the topic. The ultimate goal is to consolidate the available evidence in a way that directly addresses specific, predefined research questions, providing clear insights and guiding future research directions. By combining rigorous evaluation with comprehensive data collection, systematic and mixed review methodologies play a crucial role in advancing knowledge and informing decision-making across scientific disciplines [28].

### 2.2. Review Question and Searches Strategy

A compiled document detailing the key genes involved in PUFA biosynthesis in crops, the regulatory mechanisms governing their expression, and crop species with the highest potential for producing PUFA-rich fatty acids, is the driving force of the current review study. This question is selected due to the increasing significance of lipid research in modern living standards and the need to consolidate crop studies into a comprehensive resource for lipid metabolism researchers. Consequently, molecular research articles on crops were reviewed to evaluate the role of structured information in advancing crop genetic interventions aligned with current lipid research for functional and healthy diets. A well-defined review question is crucial, as it guides the search strategy, inclusion/exclusion criteria, and data extraction [29], while also helping readers determine the relevance of the review. Efforts were made to compile lipid studies on crops, including metabolic, biosynthesis pathways, gene regulation, and improvement in PUFA content since 2014. This review follows the PICOT framework, as outlined by Samson and Schoelles [30], to guide the selection and collection of research articles on omega-3 and omega-6 fatty acids, emphasizing crop genetic engineering and the documented health benefits of these essential fatty acids.

### 2.3. Data Sources

The search was conducted using five key databases, PubMed, Web of Science, Science Direct, Semantic Scholar, and Google Scholar, chosen for their extensive, reliable sources of biomedical and crop molecular research. Care was taken to avoid unreliable sources, which could impact the review’s conclusions. The review included only peer-reviewed, full-text quantitative studies as outlined in the search strategy.

### 2.4. Data Extraction and Software Application

Following the PICOT framework, keywords were used to conduct searches across multiple databases. The search strategy included keywords, subject headings, and MeSH headings to help broaden and capture more results as well as potential alternative terminology that could have been utilized. The following search terms were used: alpha-linoleic acid OR gamma linoleic acid OR omega-3 OR omega-6 OR docosahexaenoic acid OR eicosapentaenoic acid; genetically modified crop OR molecular breeding OR metabolic engineering plant breeding OR plant genetic manipulation OR genetic transformation OR crop biotechnology; nervous system OR cardiovascular system OR neuro-degenerative diseases OR physiological function. A total of 426 articles were retrieved from five databases (Figure 2), of which 60 were identified as duplicates. After duplicate removal, 366 articles were exported to the Zotero reference manager software. The complete Zotero file, containing 366 studies, was then imported into *Rayyan*, a tool designed for systematic screening of published articles and facilitating author collaboration, following the removal of any remaining duplicates within Zotero. The author and co-authors independently screened the articles, applying inclusion and exclusion criteria to reduce bias and errors. After successive screening, 21 original articles were selected for final study and comprehensive analysis (see summary for tabulating CASP Table 1). For the bibliometric analysis, the authors aimed to generate core information in addition to the research articles. Therefore, 73 full-text articles were assessed and subjected to VOSviewer analysis.

### 2.5. Term Map

We employed VOSviewer version 1.6.20 software to extract and analyze terms from the titles and abstracts of 73 papers [48] and visualize the findings as bubble maps. Each bubble represented a specific phrase term, and a manual view was conducted to eliminate generic or irrelevant terms [49]. The bubble size indicated the frequency of the term’s occurrence using binary counting, where multiple occurrences in a single paper were counted as one. The color of the bubbles represented the citation count for papers containing the term, with bubbles positioned closer together indicating higher co-occurrence of the corresponding terms [4,17,42,50].

## 3. Results and Discussion

### 3.1. Evidence Synthesis

The *PRISMA* flow diagram summarizes the result of the database search and articles selection process. Study characteristics were compiled using data extracted from each publication with a standardized template (Table 1). All studies reviewed were primary research, focusing on crop biotechnology advancements, particularly in crop lipid biosynthesis. The authors applied the following inclusion criteria: studies published between 2014 and 2024 years, articles written in English, relevant research areas, and primary research crop molecular studies. The exclusion criteria: systematic reviews, books, documents, conference reports, qualitative studies, non-English studies, and studies published before 2013. Finally, 21 selected studies, mostly centred on molecular research in the modification of oil crops, were considered for evidence synthesis (Figure 3).

### 3.2. Citation Counts

The citation counts of 73 papers ranged from 1 to 1157 (mean ± SD: 75.8 ± 159.5, with a cumulative total of 5454 (Appendix A). The adjusted citation count (citation per year since publication) ranged from 0 to 192.8 (mean ± SD: 18 ± 35.6, total adjusted citations = 1290.1. Saini and Keum, followed by Djuricic and Calder, authored the most cited papers on PUFAs in recent crop molecular research. The highest adjusted citation count paper focused on standardized methods for assessing PUFA progress in crops linked to essential dietary needs [51].

Highly cited articles concluded that oxidative stress and inflammation are key contributors to chronic non-communicable diseases, with PUFAs regulating antioxidant pathways, modulating inflammation, and influencing hepatic lipid metabolism and organ function, including the heart. Of the 73 most related and cited papers, 21 contained the term “crop molecular research for PUFA synthesis, mainly Omega-3, and Omega-6” in their titles, abstracts, or keywords. These studies emphasize the importance of crop research for oil production and functional foods, highlighting their translational value in nutraceuticals and modern food systems.

### 3.3. Core Journals

The analysis of the 73 selected papers was published across 53 different journals, with impact factors ranging from 1.3 to 17.2, and an average impact factor ± SD: 4.6 ± 7.02. The analysis highlights the significant contributions of the academic community, particularly researchers in the field of PUFAs and nutrition, to this area of study. Their efforts have played a crucial role in advancing research in this field are well-recognized. A statistical correlation analysis showed a significant relationship between impact factors and citation metrics. A weak negative correlation between impact factor and total citation count (r = −0.11, *p* = 0.001), suggests that journals with higher impact factors did not necessarily receive more total citations. Conversely, a weak positive correlation was found between impact factor and adjusted citation count (r = 0.11, *p* = 0.001), implying that when adjusted for certain factors, higher impact journals tended to receive more citations per article. This suggests that highly cited research on crop lipid biosynthesis and their dietary progress is often published in prestigious journals, and lipid research works are a current thematic area of study and are preferred by current researchers. Of the 53 journals, those from the United Kingdom, United States, and Switzerland published the most articles (19, 12, 11, and 11, respectively; Appendix A). This indicates that readers seeking highly cited papers on crop oil synthesis and functional foods can focus on a smaller set of prominent journals.

### 3.4. Time of Publication, Authors, and Countries/Territories

The 21 original papers on crop oil/PUFA synthesis (2014–2024) involved 148 authors from 51 institutions across 12 countries or territories. The limited number of retrieved articles is due to the use of specific search terms designed to align with the objectives of this review within the specified timeframe. Additionally, the selection focused on original research related to the molecular engineering of crop oil for PUFA synthesis. Although the field has attracted significant research interest, studies specifically addressing genetic modification in crops remain relatively scarce despite the extensive research on PUFAs.

China contributed 10 of these papers, followed by the United States with 3 (Table 2). The dominance of contributions from China and Europe aligns with their significant role in crop molecular genetic engineering research on fatty acid biosynthesis for modern healthy food innovations [52,53]. Recent conclusions highlight that Asian countries, in particular, make substantial contributions to highly cited research on nutraceuticals and functional foods derived from crop oils.

### 3.5. Term Map

A term map was created by analyzing the words in the titles and abstracts of the 73 papers, highlighting key impact terms (Figure 4). A total of 203 terms appeared in at least five papers. Papers containing certain terms had higher than average citation counts, such as human nutrition (n = 8, 119 citations per paper), oil (n = 6, 82 citations per paper), linoleic acid (n = 4, 48 citations per paper), obesity (n = 4, 43 citations per paper), gene expression (n = 3, 41 citations per paper), FA-omega-3 (n = 5, 72 citations per paper), inflammation (n = 4, 37 citations per paper), DHA (n = 3, 44 citations per paper), Arabidopsis (n = 5, 35 citations per paper), alpha-linolenic acid (n = 3, 35 citations per paper), and plants (n = 3, 40 citations per paper). These terms are mainly associated with research on crop biosynthesis of fatty acids to enhance nutritional oil quality for healthier diets. Additionally, the findings confirm earlier predictions about emerging topics related to heart health, cancer, metabolism, probiotics, and antioxidants [31]. The authors expect these topics to continue gaining importance, although antioxidants may lose relevance due to skepticism surrounding the value of simple antioxidant assays in food product studies [43,46].

### 3.6. Appraising the Evidence of 21 Original Research Articles and Summary

Systematic and bibliometric review evaluations must assess studies for quality and methodological rigor, but this process differs between quantitative and qualitative research due to variation in procedures, sample sizes, and sampling strategies [55]. Researchers often use the Critical Appraisal Skills Programme (CASP) method to evaluate the quality of health and related evidence synthesis. Data from each article were extracted using a standardized tool and used to create a key Characteristics Table (Table 1).

For example, Fan with his colleagues cloned the *FAD2* gene in *Idesia polycarpa*, which significantly enhanced linoleic acid synthesis [32]. Similarly, the overexpression of the *PvFAD3* gene in tobacco resulted in a notable increase in ALA synthesis, indicating its important role in improving oilseed crops [19]. The *WRI1* from *Arabidopsis thaliana* and potato was expressed in *Nicotiana benthamiana*, showing high expression during seed and embryo development.

Overall, various authors have identified and screened genes contributing significantly to crop metabolism and lipid biosynthesis production, including *TMT* (Soybean), *HGGT* (Barley) [31], *PvFAD3 (Plukenetia volubilis*) [33], *WRI1 (A. thaliana*, Potato, Oat, and Nutsedge) [19], *GhKAR*; *GhHAD; GhENR; GhFAD2* (*Gossypium hirsutum*) [38], etc. (see Table 1). All research findings can be summarized as the current crop improvement strategies primarily aim to enhance PUFA content by overexpressing key *FAD* genes, including *FAD2*, *FAD3*, *FAD6*, *FAD7*, and *FAD8* and related homologous genes, and key transcription factors. In addition, some research articles indicates that PUFA levels in crops act as key indicators of cell maturity and aging, emphasizing the vital role of lipid synthesis in plant growth and development.

### 3.7. The Broad Spectrum of PUFAs: ALA and LA Oil Crop Biosynthesis and Progression (2014–2024)

In line with the joint FAO/WHO expert consultation on fats and fatty acids in human nutrition [56], the genetic modifications have enhanced PUFA content in plants particularly oil crops, as evidenced by publications listed in Table 2. As shown in Table 2, many of the most cited papers on crop PUFA biosynthesis were published in journals such as Life Science, Nutrients, Lipid in Health and Disease, International Journal of Molecular Sciences, BMC Plant Biology, Molecular Plant, Frontiers in Oncology, and Plant physiology. Numerous studies focused on crop lipid biosynthesis, a foundational topic in the field of dietary oils like DHA and EPA, which are essential in modern nutrition.

These findings suggest that the study of essential fatty acid synthesis in crops spans a wide spectrum, intersecting crop molecular research with food science, nutrition, and healthcare. Manipulating plant lipid composition enables the development of novel oil-producing crops. Specifically, crop essential fatty acids synthesis is rooted in lipid biosynthesis and serves as a critical link between food science engineering and nutrition, contributing to a healthier food system (Table 3).

### 3.8. Profiling Naturally Oil Rich Crops and Schematic Representation of Lipid Biosynthetic Pathway

Various oil crop species and their cultivars exhibit significant variation in both the content and composition of fatty acid stored in their seeds, without notable differences in plant physiology (Table 2). Fatty acids are initially generated in the plastid and are transported to the endoplasmic reticulum for TAG synthesis (Figure 5). Plant-based sources of essential fatty acids are now seen as the primary alternative to meet growing market demand because of International ban on shark fishing and for sustainable environment [59].

### 3.9. Study Limitations

While systematic and bibliometric reviews offer valuable insights, their limitations must be acknowledged. One major limitation is the reliance on specific databases, which may exclude relevant studies published elsewhere. Additionally, unpublished studies, theses, and industry reports are often difficult to include, leading to potential gaps in analysis. Non-English research and regional studies may be underrepresented in global databases, limiting the review’s comprehensiveness. Additionally, restricting document collection to 2014–2024 may exclude relevant earlier studies.

## 4. Conclusions and Future Directions

Recent research aims to optimize crop genetic modifications to improve PUFA yield and stability for large-scale production. This review study examines advancements in PUFA biosynthesis in genetically modified crops, providing an in-depth analysis of existing literature. It highlights opportunities for further research to enhance understanding of dietary shifts toward higher PUFA intake to reduce diet-related health issues. This review employs a systematic and mixed-methods approach to analyze 73 highly relevant and widely cited papers on crop lipid biosynthesis, with a particular focus on VLPUFA-related studies. The top-ranked studies addressed the role of lipid biosynthesis and dietary oil in promoting health. Essential food lipids, such as ALA and LA-enriched oils, have significant health benefits and are widely used as nutritional supplements.

Key genes involved in crop seed oil production *ACCase*, DGAT, and others like *FAD2*, *FAD3*, *TMT*, *HGGT*, *GhKAR*, *GhHAD*, *GhENR,* and transcription factors of *MYB89*, *MYB96*, *WRI*, *LEC*, *GL2*, *FUS3,* and *HB2*—are involved in lipid biosynthesis to drive fatty acid biosynthesis and TAG accumulation. Beyond traditional oil crops, research is exploring novel oil sources such as vegetative tissues (leaves and stems) and microalgae, which offer significant potential for oil production.

Over the past decades, efforts have focused on enhancing oil content and producing value-added oils, such as PUFA, however poor transgene expression of new metabolites, which may disrupt cellular processes. Further research is needed to understand the mechanisms of lipid biosynthesis, transport, storage, and regulation. Precision nutrition, which tailors lipid composition to individual health needs, is a promising direction for future functional lipid development. As the saying goes, “Let food be thy medicine and medicine be thy food”.

## Figures and Tables

**Figure 1 ijms-26-03462-f001:**
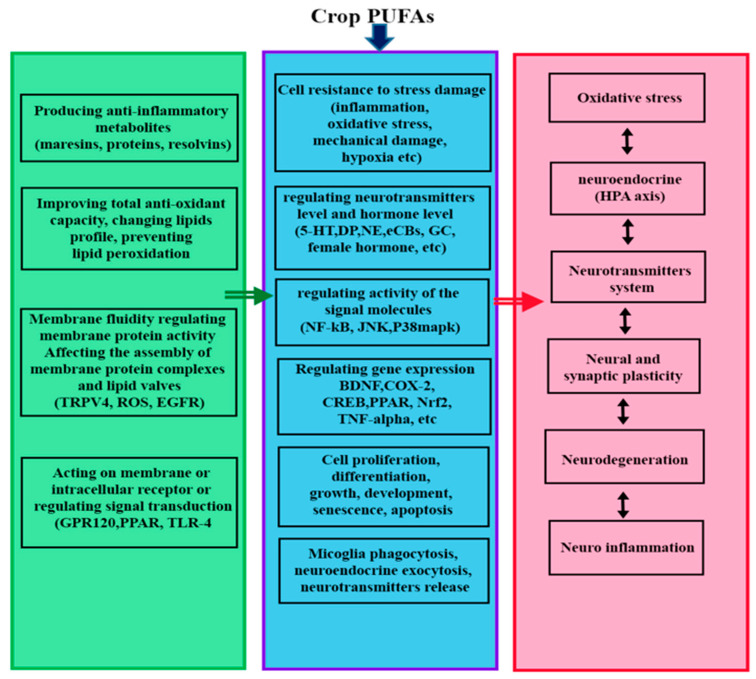
The health benefits of PUFAs.

**Figure 2 ijms-26-03462-f002:**
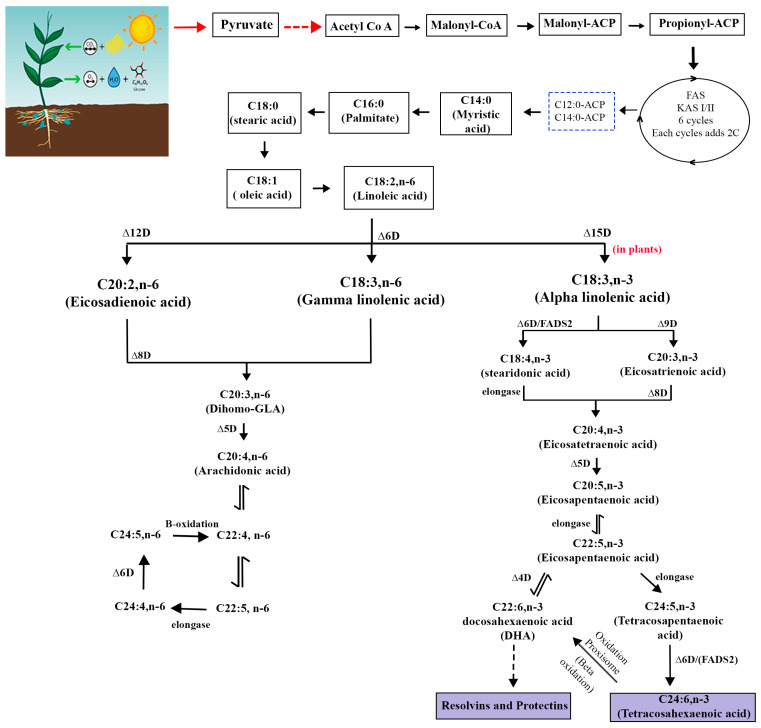
The biosynthetic pathways of PUFAs including the enzymes mentioned in crops.

**Figure 3 ijms-26-03462-f003:**
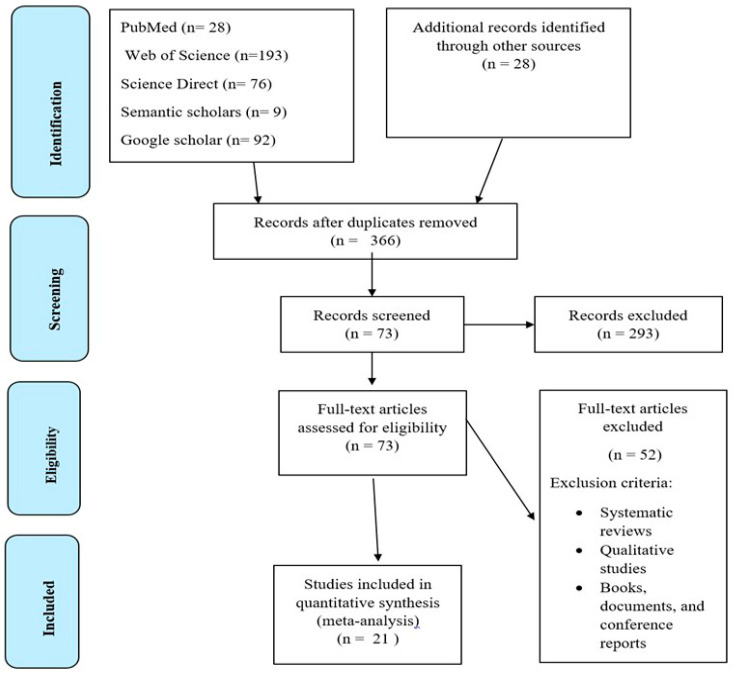
PRISMA flow diagram.

**Figure 4 ijms-26-03462-f004:**
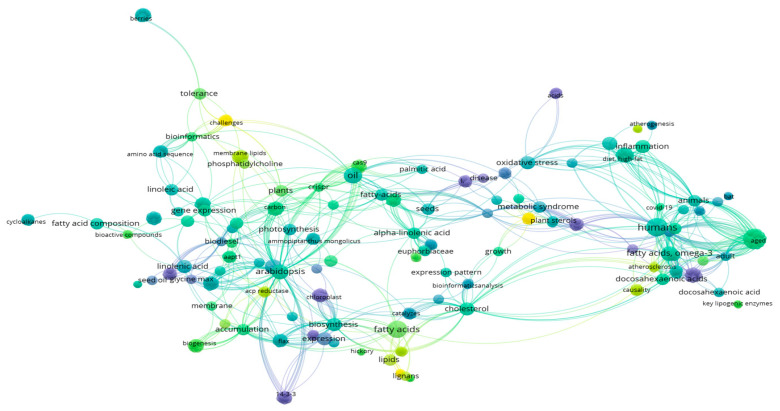
Term map using words from titles and abstracts of the 73 most popular crop molecular findings on PUFAs: omega 3/6 biosynthesis for current dietary food values papers. Words from titles and abstracts were parsed, analyzed, and visualized by VOS viewer. There were 203 terms that appeared in five or more papers and hence included in the map. Each bubble represents a term or phrase. The bubble size indicates its frequency of occurrence. The bubble color indicates the averaged citation counts received by papers containing the term or phrase. If two terms co-occur more frequently, the two bubbles are in closer proximity.

**Figure 5 ijms-26-03462-f005:**
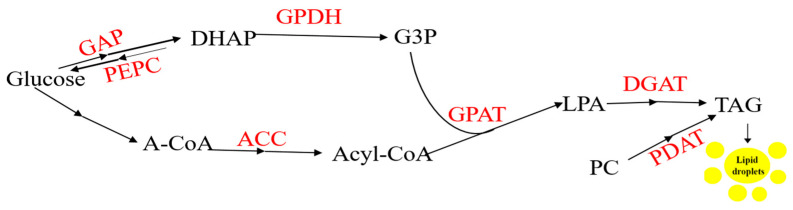
Schematic representation of lipid biosynthetic pathway in plants and major genes involved. GAPC, glyceraldehyde-3-phosphate dehydrogenase; PEPC, phosphoenolpyruvate carboxylase; DHAP, dihydroxyacetone phosphate; GPDH, glycerol-3-phosphate dehydrogenase; G3P, glycerol-3-phosphate; A-CoA, acetyl-CoA; ACC, acetyl-CoA carboxylase; GPAT, glycerol-3-phosphate acyltransferase; LPA, lysophosphatidic acid; DGAT, diacylglycerol acyltransferase; PC, phatidylcholine; PDAT, phospholipid diacylglycerol acyltransferase; TAG, triacylglycerol(lipid droplets).

**Table 1 ijms-26-03462-t001:** Characteristics of studies most 21 original articles for crop PUFA biosynthesis research findings.

Authors/Year	Titles	Genes/Genotypes/Varieties Used for Study)	Measurements	Findings/Conclusions
[31], USA/China	Metabolic engineering of soybean seeds for enhanced vitamin E tocochromanol content and effects on oil antioxidant properties in polyunsaturated fatty acid-rich germplasm	*Soybean* line (SDA-1/535-9); *γTMT* gene, *HGGT* (Barley) gene	Oil contents of transgenic soybean seeds oi, transgenic *HGGT* barley gene	Soybean line crossed—*γTMT* showed oil enriched in SDA, ALA, GLA; the progeny of *HGGT* expressed had ≥6 fold increased free radicals scavenging activity in cell metabolic activity
[32], China	Molecular cloning and function analysis of *FAD2* gene in Idesia polycarpa	*FADS2* gene, *Indesia polycarpa* fruit tree	Lipid accumulation, linoleic accumulation rates and final linoleic content	*IpFAD2* gene could encode a bio-functional omega-6 fatty acid desaturase; *IpFAD2* has significant contribution in linoleic synthesis
[33], China	Overexpression of *PvFAD3* gene from Plukenetia volubilis promotes the biosynthesis of linolenic acid in transgenic tobacco seeds	Gene *PvFAD3*, *Nicotiana benthamiana* (for transformation and test material)	Expression of *PvFAD3* gene in different tissues *of P. volubilis.*	The transgenic seed showed a significant increase in-ALA content, a dramatic decrease in LA content; *PvFAD3* gene of was confirmed as a key enzyme gene for ALA synthesis
[34], China	De novo transcriptome assembly of the eight major organs of Sacha Inchi (*Plukenetia volubilis*) and the identification of genes involved in α-linolenic acid metabolism	Major organs of Sacha: roots, stems, shoot apexes, mature leaves, male flowers, female flowers, fruits, and seeds	Expression of genes related to the ALA metabolism based on the de novo assembly and annotation transcriptome	*Sacha Inchi* accumulates high level of ALA in seeds by strong expression of biosynthesis-related genes, the upregulation of *FAD3* and *FAD7* is consistent with high level of ALA in seeds of *Sacha Inchi*
[19], Sweden	Transcriptional transitions in *Nicotiana benthamiana* leaves upon induction of oil synthesis by WRI1 homologs from diverse species and tissues	*WRI1* (*A. thaliana*/potato/poplar/oat) were expressed in *Nicotiana benthamiana*	Oil synthesis by *WRI1*	Transcripts representing fatty acid degradation were upregulated indicating that fatty acids might be degraded to feed the increased need to channel carbons into fatty acid synthesis creating a futile cycle. WRI1 may exert on global gene expression during seed and embryo development
[35], USA/Philippines	Fatty acids, triterpenes, and cycloalkanes in ficus seed oils	Ficus spp: *F. nota*, *F. septica*, *F. ulmifolia)*	Level of FAs studied in various plant parts	ALA is the most prominent FA in the seed oils followed by LA, with these two fatty acid comprising about 75% of the fatty acids in the oils
[36], Denmark	Sacha Inchi (*Plukenetia volubilis* L.) is an underutilized crop with a great potential	*P. volubilis*	FA contents studied	The seed contains 35.2–50.8% ALA and 33.4–41.0% LA
[37], France	Oleic conversion effect on the tocopherol and phytosterol contents in sunflower oil	Hundreds of hybrids and parental lines	FA contents studied	The results indicated that sunflower oil is rich in *α-tocopherol* and *β-sitosterol*
[38], China	Genetic and morpho-physiological differences among transgenic and no-transgenic cotton cultivars	*G. hirsutum* genotypes namely: *GhKAR*, *GhHAD*, and *GhENR*, and cultivars (10H1014, 10H1041, 10H1007 and 2074)	Expression patterns of genotypes and cultivars were studied	Oil contents of *GhKAR* and *GhENR* overexpression lines increased 1.05~1.08 folds; these results indicated that *GhHAD*, *GhENR*, and *GhKAR* were involved in both seed oil synthesis
[39], Spain	Specialized functions of olive *FAD2* gene family members related to fruit development and the abiotic stress response	Three cDNA sequences: *OepFAD2-3*, *OepFAD2-4 OepFAD2-5* all encoding three microsomal *FAD2* gene from olive *(Olea europaea cv. Picual)*	Three cDNA genes expression and lipid analysis	*OeFAD2-5*, together with *OeFAD2-2* contributes mostly to the linoleic acid present in the mesocarp and, therefore, in the olive oil
[40], China	Constitutive expression of chloroplast G3P acyltransferase from *Ammopiptanthus mongolicus* enhances unsaturation of chloroplast lipids and tolerance to chilling, freezing and oxidative stress in transgenic *Arabidopsis*	*AmGPAT*, *Arabidopsis*	Characterize the physiological function of AmGPAT from *A. mongolicus*	Transgenic lines of AmGPAT in *Arabidopsis* increased the levels of cis-unsaturated fatty acids: ALA
[41], China	Combined genome-wide association analysis and transcriptome sequencing to identify candidate genes for flax seed fatty acid metabolism	Flax seeds (224 samples)	Based on GWAS and RNA-seq methods to identify candidate genes for fatty acid metabolism in flax seeds	Among 10 candidate genes screened, 2 most genes were significantly correlated with 5 fatty acid contents in seeds of the high oil variety: (Shuangya4vs.NEW), and both of these genes encode acyl-lipid *omega-3 desaturase*
[42], China	*GhFAD2–3* is required for another development in *Gossypium hirsutisms*	Cotton *GhFAD2* gene family	Molecular characterization of *GhFAD2 gene*	The ratio of monounsaturated to polyunsaturated fatty acid was 5.43 in *fad2–3* anther, which was much higher than that of the WT (only 0.39)
[17],United States	CRISPR/Cas9-induced *fad2* and *rod1* mutations stacked with *fae1* confer high oleic acid seed oil in pennycress (*Thlaspi arvense* L.)	Genes *FAD2* and *ROD1*	Knockout mutation	*fad2*, *fae1* and *rod1* fae1 double mutants produced 90% and 60% oleic acid in seed oil, respectively, with PUFAs in *fad2 fae1* as well as *fad2* single mutants reduced to less than 5%
[43], India	Food and nutraceutical functions of sesame oil: an underutilized crop for nutritional and health benefits	Sesame crop	Sesame crop demonstration	Potential candidate to maintain the diversity of food oils
[44], China	Yacon (*Smallanthus sonchifolius*) tuber: a novel and promising feedstock for enhanced high-value docosahexaenoic acid production by *Schizochytrium* sp.	Yacon tuber hydrolysate (YTH)	Demonstration	*YTH* is a novel potential substrate for *Schizochytrium* sp. ATCC 20888 to produce DHA
[45], China/Switzerland	Metabolomics in combination with network pharmacology reveals the potential anti-neuro inflammatory mechanism of essential oils from four Curcuma species	Four Curcuma species: *Curcuma longa* L. (CL), *Curcuma kwangsiensis* S.G. Lee and C.F. Liang (CK), *Curcuma zedoaria* (Christm.) *Roscoe* (CZ) and *Curcuma aromatica Salisb*. (CA)	Demonstration	11 (e.g., ar-turmerone, curzerenone, nootkatone, curlone, β-Elemene, curzerene) metabolites mainly associated with arachidonic acid metabolism, linoleic acid metabolism, and aminoacyl-tRNA biosynthesis were identified
[46], Iran	Genetic structure and diversity of Iranian Cannabis populations based on phytochemical, agro-morphological and molecular markers	Hemp (*Cannabis sativa*), 10 local cultivated landraces	Diversity based seed oil fatty and biochemical traits	The studied populations showed that *Kermanshah* had the highest proportion of linoleic acid (54.03%), *Hamedan*; α-linolenic acid (21.47%), Alborz; oleic acid (16.75%), Kohkylouih-and-Boyerahmad; palmitic acid (6.82%), and Khuzestan; stearic acid (3.23%)
[1], India	Exploring lipid health indices and protein quality in ninety Indian linseed varieties by comprehensive analysis of fatty acid composition, lignan content, and amino acid composition	90 linseed varieties	Diversity based seed oil fatty and biochemical traits	The analysis FA composition revealed 5 major FAs as ALA, LA, oleic acid (OA), palmitic acid (PA), and stearic acid (SA) ranging from 1.68% to 59.19%, 5.61–62.44%, 18.14–45.54%, 5.69–9.5%, and 3.94–10.29%, respectively
[2], China	Analysis of lipidomics profile of *Carya cathayensis* nuts and lipid dynamic changes during embryonic development	Hickory nuts	Lipid omics profiling and analysis	544 kinds of lipids were identified in mature hickory nuts: TAG, DAG, and other related lipids had high relative content with abundance of unsaturated fatty acids, such as oleic acid, linoleic acid and linolenic acid, localized mainly at sn-2 lipid position.
[47], Italy/France	Camelina [*Camelina sativa* (L.) Crantz] seeds as a multi-purpose feedstock for bio-based applications	Six *camelina* cultivars (Cypress, Midas, 789-02, Pearl, Omega, and WUR)	Profiling *Camelina* seeds for FA contents and related lipids	Pearl and 789-02 were identified as the most suitable for specific bio-based applications because of the increased omega-3 to omega-6 ratio of the oil. Pearl represent a starting point for future research targeting the increase/decrease of specific fatty acids

**Table 2 ijms-26-03462-t002:** Total LA and ALA contents of oils and foods commonly consumed the years 2024/2025.

Source	Unit (g)	18:2n-6	18:3n-3	References
Canola	100 g	18.64	9.137	[6,54]
Flaxseed/linseed	100 g	14.25	53.37	
Soybean	100 g	50.42	6.789	
Walnut	100 g	52.90	10.40	
Sunflower	100 g	3.61	0.192	
High oleic safflower	100 g	12.72	0.096	
Almonds, raw	100 g	12.30	0.003	
Amaranth	100 g	2.736	0.042	
Avocados	100 g	1.674	0.111	
Brazil nuts	100 g	23.859	0.018	
Cashews, raw	100 g	7.782	0.062	
Chia seeds	100 g	5.840	17.80	
Hempseed	100 g	1.340	8.864	
Millet, cooked	100 g	0.480	0.028	
Oat bran, cooked	100 g	0.324	0.015	
Pistachio, raw	100 g	13.10	0.210	
Poppy seeds	100 g	28.30	0.273	
Quinoa	100 g	2.977	0.260	
Rye	100 g	0.659	0.108	
Sesame seeds	100 g	21.375	0.376	
Aracauriaceae oil	100 g	36.9	11	
Boraginase oil	100 g	22	21.6	
Echium oil	100 g	19.5	28.1	
Blackcurrant oil	100 g	40–50	12–15	
Evening promise	100 g	70–77	0.1–1	
Primulaceae	100 g	33.1	20.8	
Rapeseed oil	100 g	21.6	7.3–11.1	[5]
Corn oil	100 g	59.0	1.16	
Mustard seed oil	100 g	15.0	6.0	

**Table 3 ijms-26-03462-t003:** Molecular research on crop lipid biosynthesis.

Genus	FA Modified	Target Enzyme Engineered	Sources of Genes	Molecular Technique Applied	Reference
*B. napus*	Palmitic acid	*Acyl-ACP thioesterase*	*Cuphea*	OE	[57]
	Lauric acid	*Acyl-ACP thioesterase*	*California Bay*	OE	
	Caprylic acid	*Acyl-ACP thioesterase*	*Cuphea*	OE	
	Stearic acid	*Acyl-ACP thioesterase* *Laurate-specific* *Lysophosphatidic acid acyl-transferase*	*Mangosteen* *Coconut*	OE	
	Oleic acid	*Oleoyl-Δ-12-desaturase*	*B. napus*	DR	
	GLA(18:3ω-6)	*Oleoyl-Δ-6-desaturase and Oleoyl-Δ-12-desaturase*	*Mortierellaapina*	OE	
	Saturated fatty acids	*Palmitoyl-ACP-desaturase*	*Doxanthaunguis cati*	DR	
	Oil rich in ricinoleic acid	*Fatty acid hydroxylase*	*Castor*	OE	
*B.juncea*	Oleic acid	*Oleoyl-Δ-12-desaturase*	*Brassica*	DR	
Cotton seed	Stearic acid	*Stearoyl-ACP-Δ-9-desaturase*	*Cotton seed*	DR	[38]
	Oleic acid	*Oleoyl-Δ-12-desaturase*	*Cotton seed*	DR	
Flax	C18 omega-3, stearidonic acid	*omega-3 desaturase*	*Primula*	OE	[41]
Soybean	Stearic acid	*Stearoyl-ACP-Δ-9-desaturaseand oleoyl-Δ-12-desaturase*	*Soybean*	DR	[58]
	Oleic acid	*Oleoyl-Δ-12-desaturase*	*Soybean*	DR	
	Eleostearic acid	*Conjugase*	*Momordica*	OE	
	Δ-5 Eicosenoic acid	*β-Ketoacyl-CoAsynthase andacyl-CoAdesaturase*	*Meadowfoam*	OE	
	Oleic acid	*Δ-12Fatty acid desaturase*	*Soybean*	OE	
	Palmitic acid	*Palmitoyl-thioesterase*	*Soybean*	DR	
	GLA and stearidonic acid	*Δ-6Desaturase*	*Arabidopsis thaliana*	OE	
	Arachidonic acid	*Δ-5Desaturase*, *Δ-6desaturase*, *GLELO elongase*, *Δ-15desaturase*	*Mortierella alpine and soybean*	OE	
	Stearidonic acid	*Δ-6Desaturase*, *Δ-15desaturase*	*Brassica officinalis and A. thaliana*	OE	
	C20-LCPUFAs	*Δ-6Desaturase*, *Δ-6elongase*, *Δ-5 desaturase*	*Marchantiapolymorpha*	OE	
	Improved oil yield and composition	*Fatty acid ω-3 destaurase2*, *ACP thioesterase*, *diacylglycerol acyl-transferase*, *dihydrodipicolinate synthase*, *high-lysineprotein*, *truncatedcysteine synthase*	*Soybean, Yarrowia lipolytica*, *Corynebacteriumglutamicum*, *barely*	OE	
	Increase oil content	*Sphingo lipid compensation*	*Saccharomycescerevisiae*	OE	
	Increase oil content	*Diacylglycerolacyl-transferase2A*	*Umbelopsisramanniana*	OE	
	Epoxy fatty acid	*Epoxygenase*, *diacylglycerolacyl transferase*	*StokesialaevisVernonia galamensis*	OE	

Note: OE, overexpression; DR, downregulation.

## Data Availability

The original contributions presented in the study are included in the article; further inquiries can be directed to the corresponding author(s).

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
