# Peer review of "Advancements in Crop PUFAs Biosynthesis and Genetic Engineering: A Systematic and Mixed Review System"

_ijms, 2025, doi:10.3390/ijms26083462_

Round 1
Reviewer 1 Report
Comments and Suggestions for Authors
This manuscript by Assefa et al. summarizes recent key information on the biosynthesis and genetic engineering of PUFAs in crops. The study was generally well-designed, and the manuscript is well-written and structured. The following comments do not influence my positive impressions of the quality of the manuscript.
- Add the definition of abbreviations in figure and table legends.
- “Essential food lipids, such as ALA and LA-enriched oils, have significant health benefits and are widely used as nutritional supplements.” The abstract, introduction, and conclusion highlight the importance of PUFAs for human health. However, the health attributes of PUFAs were not reviewed.
- Could you add the review of the recent knowledge of the molecular regulation of PUFA biosynthesis and accumulation and factors that influence the variation of PUFA content in crops?
- Revise minor typographic errors.
Author Response
Reviewers' comments:
Reviewer #1:
This manuscript by Assefa et al. summarizes recent key information on the biosynthesis and genetic engineering of PUFAs in crops. The study was generally well-designed, and the manuscript is well-written and structured.
> Recommendation: Revise minor typographic errors.
The following comments do not influence my positive impressions of the quality of the manuscript.
Comment #1: Add the definition of abbreviations in figure and table legends.
Response: Thank you so much for valuable suggestion. We have corrected it.
Comment #2: “Essential food lipids, such as ALA and LA-enriched oils, have significant health benefits and are widely used as nutritional supplements.” The abstract, introduction, and conclusion highlight the importance of PUFAs for human health. However, the health attributes of PUFAs were not reviewed.
Response: Thank you so much. We have added necessary information as per your suggestion in revised manuscript.
Comment #3: Could you add the review of the recent knowledge of the molecular regulation of PUFA biosynthesis and accumulation and factors that influence the variation of PUFA content in crops?
Response: Thank you so much for such a detailed suggestion; we have thoroughly addressed your comment.

Reviewer 2 Report
Comments and Suggestions for Authors
Assefa et al. aimed to systematically summarize the research progress in Crop PUFAs Biosynthesis and Genetic Engineering by combining a literature review with bibliometric methods. This topic could provide essential knowledge references for researchers in related fields and holds practical significance. However, the authors' description of the literature retrieval process is vague, and the criteria for literature screening are unclear, which affects the reproducibility of this study and makes it difficult for other researchers to judge the accuracy of the results. Additionally, the manuscript has several other issues, such as incomplete citation of references, unclear logical flow, and a significant amount of meaningless analysis. Specific comments are as follows:
- The introduction lacks sufficient references (Lines 30-65; Lines 114-129).
- The subheading "2.1. Introduction" is confusing.
- In Part 2.2, "Review Question and Search Strategy," the authors state: "The review topic was chosen due to the researcher’s interest in recent molecular evidence linking crop essential oil to a healthy diet. A well-defined review question is crucial, as it guides the search strategy, inclusion/exclusion criteria, and data extraction [13], while also helping readers determine the relevance of the review. Articles from 2014 to 2024 were retrieved from five major databases" (Lines 130-135). What do the authors mean by "The review topic was chosen due to the researcher’s interest in recent molecular evidence linking crop essential oil to a healthy diet"? Additionally, the sentence "Articles from 2014 to 2024 were retrieved from five major databases" appears unrelated to the preceding content in this section.
- What retrieval method did the authors use? Keyword search? Topic search? What do the authors mean by "A total of 366 articles (after removing duplicates) 73 were selected for full-text review" (Line 145)? What caused such a large number of duplicate articles? Could it be due to an inappropriate retrieval method? I attempted a topic search on Web of Science (WOS) using several keywords and easily found target articles,please see: 【(polyunsaturated fatty acid)AND ((Plant)OR (Crop))AND (Biosynthesis)https://webofscience.clarivate.cn/wos/woscc/summary/35d481d9-18fb-4ebf-ba8f-c41057eb38f0-0149caff87/relevance/1】
- "Finally, 21 selected studies, most centred on molecular research in the modification of oil crops, were published in English between 2014 and 2024" (Lines 167-168). The authors should provide clear inclusion and exclusion criteria rather than relying on subjective preferences; otherwise, the results will have limited reference value for other researchers.
- "...confirming our hypothesis" (Lines 185-194). What is the authors' hypothesis? "This suggests that highly cited research on crop lipid biosynthesis and their dietary progress is often published in prestigious journals. ... This indicates that readers seeking highly cited papers on crop oil synthesis and functional foods can focus on a smaller set of prominent journals." What is the significance of such analysis?
- "The 21 papers included in quantitative synthesis were published between 2014–2024..." (Line 196). The authors have already conducted a literature search within the timeframe of 2014–2024. Could these articles possibly come from any other time period?
- The authors ultimately selected 21 papers (Line 167). Why are 21 papers used for some analyses while 73 papers are used for others? What is the significance of the 21 papers selected by the authors?
- How did the authors conclude that "...Numerous studies focused on crop lipid biosynthesis, a foundational topic in the field of dietary oils like DHA and EPA, which are essential in modern nutrition" (Lines 239-243) based on the journals in which the papers were published?
- "Specifically, crop essential fatty acids synthesis is rooted in lipid biosynthesis and serves as a critical link between food science engineering and nutrition, contributing to a healthier food system" (Line 247). What is the evidence for the claim that it "...serves as a critical link between..."?
- Figure 3 does not show any connecting lines.
- "Plant-based sources of essential fatty acids are now seen as the primary alternative to meet growing market demand because of the international ban on shark fishing" (Line 267). What is the basis for this claim by the authors?
- Some journal information in Table 3 may be incorrect (e.g., Plant Biotechnology).
- Table 4 lacks units for some metrics.
- "3.9. Study Limitations: One limitation of our study is that publications were extensively sourced from databases such as PubMed, Web of Science, Semantic Scholar, and Google Scholar. These databases were selected due to their broad coverage of biomedical and crop biotechnology literature. A second limitation is that the documents were collected only within the time frame from 2014 to 2024, potentially restricting the scope of earlier relevant research" (Lines 279-284). These limitations could clearly be addressed through adjustments in the search methodology. Why do the authors consider these aspects as limitations, and why did they proceed with this approach for literature retrieval and analysis?
Author Response
Reviewer #2
Assefa et al. aimed to systematically summarize the research progress in Crop PUFAs Biosynthesis and Genetic Engineering by combining a literature review with bibliometric methods. This topic could provide essential knowledge references for researchers in related fields and holds practical significance. However, the authors' description of the literature retrieval process is vague, and the criteria for literature screening are unclear, which affects the reproducibility of this study and makes it difficult for other researchers to judge the accuracy of the results. Additionally, the manuscript has several other issues, such as incomplete citation of references, unclear logical flow, and a significant amount of meaningless analysis.
Specific comments are as follows:
Comment #1: The introduction lacks sufficient references (Lines 30-65; Lines 114-129).
Response: Thank you so much for valuable suggestion. We have corrected it.
Comment #2: The subheading "2.1. Introduction" is confusing.
Response: Thank you so much for such a detailed suggestion; we have thoroughly revised the subtitle 2.1.
Comment #3: In Part 2.2, "Review Question and Search Strategy," the authors state: "The review topic was chosen due to the researcher’s interest in recent molecular evidence linking crop essential oil to a healthy diet. A well-defined review question is crucial, as it guides the search strategy, inclusion/exclusion criteria, and data extraction [13], while also helping readers determine the relevance of the review. Articles from 2014 to 2024 were retrieved from five major databases" (Lines 130-135). What do the authors mean by "The review topic was chosen due to the researcher’s interest in recent molecular evidence linking crop essential oil to a healthy diet"? Additionally, the sentence "Articles from 2014 to 2024 were retrieved from five major databases" appears unrelated to the preceding content in this section.
Response: Thank you for highlighting this. The entire Subtitle 2.2 has been modified.
Comment #4: What retrieval method did the authors use? Keyword search? Topic search? What do the authors mean by "A total of 366 articles (after removing duplicates) 73 were selected for full-text review" (Line 145)? What caused such a large number of duplicate articles? Could it be due to an inappropriate retrieval method? I attempted a topic search on Web of Science (WOS) using several keywords and easily found target articles,please see: 【(polyunsaturated fatty acid)AND ((Plant)OR (Crop))AND (Biosynthesis)https://webofscience.clarivate.cn/wos/woscc/summary/35d481d9-18fb-4ebf-ba8f-c41057eb38f0-0149caff87/relevance/1】
Response: Thank you so much for your valuable suggestion. We have revised the section according to the reviewer feedback. Following table explained out retrieving data strategy.
PICOT, authors used the following searching strategies
|
Target Population
|
Intervention or Issue |
Comparison (or None) |
Outcome |
Time (as appropriate)
|
OR Alternative Term
|
Alpha linoleic acid OR ALA OR Gama linoleic acid OR GLA OR Omega-3 OR ω-3 OR ω -6 OR Omega-6 OR Omega 3/6 OR OR Docosahexaenoic acid OR DHA OR Eicosapentaenoic acid OR EPA |
GMO OR Genetically modified crop OR Molecular breeding OR Genetically modified organism OR metabolic engineering OR genetic engineering OR plant breeding OR Plant genetic manipulation OR Genetic transformation OR Crop biotechnology |
|
Nervous system OR CNS OR CVS OR cardiovascular system OR BP OR Blood pressure OR Brain development OR neuro-degenerative diseases OR Cholesterol OR Alzheimer’s disease OR dementia OR physiological function |
2014 – 2024 |
PubMed search
Google scholar search:
“Alpha linoleic acid” OR “ALA” OR “Gamma linoleic acid” OR “GLA” OR “Omega-3” OR “ω-3” OR “ω -6” OR “Omega-6” OR “Omega 3/6” OR “Docosahexaenoic acid” OR “DHA” OR “Eicosapentaenoic acid” OR “EPA” OR “GMO” OR “Genetically modified crop” OR “Molecular breeding” OR “Genetically modified organism” OR “metabolic engineering” OR “genetic engineering” OR “plant breeding” OR “Plant genetic manipulation” OR “Genetic transformation” OR “Crop biotechnology” OR “Nervous system” OR “CNS” OR “CVS” OR “cardiovascular system” OR “BP” OR “Blood pressure” OR “Brain development” OR “neuro-degenerative diseases” OR “Cholesterol” OR “Alzheimer’s disease” OR “dementia” OR “physiological function”=92
Wos
(((((((((((((((((((((((ALL=(Alpha linoleic acid)) OR ALL=(Gamma linoleic acid)) OR ALL=(Omega-3)) OR ALL=(Omega-6)) OR ALL=(Docosahexaenoic acid)) OR ALL=(Eicosapentaenoic acid)) AND ALL=(Genetically modified crop)) OR ALL=(Molecular crop breeding)) OR ALL=(Genetically modified organism)) OR ALL=(metabolic engineering)) OR ALL=(genetic engineering)) OR ALL=(plant breeding)) OR ALL=(Plant genetic manipulation)) OR ALL=(Genetic transformation)) OR ALL=(Crop biotechnology)) AND ALL=(Nervous system)) OR ALL=(cardiovascular system)) OR ALL=(Blood pressure)) OR ALL=(Brain development)) OR ALL=(neuro-degenerative diseases)) OR ALL=(Cholesterol)) OR ALL=(Alzheimer’s disease)) OR ALL=(dementia)) OR ALL=(physiological function)=193
SCIENCE DIRECT SEARCH:
(‘’Alpha linoleic acid’’ OR ‘’Gamma linoleic acid’’ OR ‘’Docosahexaenoic acid’’ OR ‘’Eicosapentaenoic acid’’) AND (‘’Molecular breeding’’ OR ‘’genetically modified organism’’ OR ‘’Crop biotechnology’’) AND (“Nervous system “OR “Alzheimer’s disease”)
Problem/Issue: Does the Current crop molecular research evidence linking PUFAs: Omega 3/6 in modern society dietary essentials and how about the progress for better health? |
Search Time Frame: 2014 to 2024 |
Selected databases & sources: PubMed: 28 articles retrieved. Saved 2 on Zotero Web of Science: 193 articles retrieved. Saved 2 on Zotero Science Direct: 76 articles retrieved. Saved 2 on Zotero Semantic scholars: 9articles retrieved. Saved 2 on Zotero Google scholar: 92 articles retrieved. Saved 2 on Zotero Additional records identified through other sources: 28 articles retrieved. Saved 2 on Zotero. |
Inclusion criteria: · Studies published from 2014, to 2024 · English language only. · Research areas · Primary research works Exclusion criteria: · Systematic reviews · Qualitative studies · Books, documents, and conference reports · Studies not in English · Studies before 2013 |
Comment #5: "Finally, 21 selected studies, most centred on molecular research in the modification of oil crops, were published in English between 2014 and 2024" (Lines 167-168). The authors should provide clear inclusion and exclusion criteria rather than relying on subjective preferences; otherwise, the results will have limited reference value for other researchers.
Response: Thank you so much for your valuable suggestion. We have added inclusion and exclusion criteria in revised manuscript (Lines 201- 205).
Comment #6: "...confirming our hypothesis" (Lines 185-194). What is the authors' hypothesis? "This suggests that highly cited research on crop lipid biosynthesis and their dietary progress is often published in prestigious journals. ... This indicates that readers seeking highly cited papers on crop oil synthesis and functional foods can focus on a smaller set of prominent journals." What is the significance of such analysis?
Response: We greatly appreciate your valuable feedback. We have made the necessary corrections and rewritten the analysis, from Line 224 to line 236.
Comment #7: "The 21 papers included in quantitative synthesis were published between 2014–2024..." (Line 196). The authors have already conducted a literature search within the timeframe of 2014–2024. Could these articles possibly come from any other time?
Response: Thank you very much. Although research on crop PUFAs exists, we excluded older studies, focusing specifically on recent data from 2014 to 2024 to ensure the relevance of our analysis.
Comment #8: The authors ultimately selected 21 papers (Line 167). Why are 21 papers used for some analyses while 73 papers are used for others? What is the significance of the 21 papers selected by the authors?
Response: Thank you for your valuable feedback. Since this study follows systematic and mixed review approaches, we applied both systematic and bibliometric review methods. Accordingly, 21 articles were utilized to generate original information, while 73 articles were included for bibliometric analysis. This approach was intentionally adopted to enhance the core information beyond the original research articles.
Comment #9: How did the authors conclude that "...Numerous studies focused on crop lipid biosynthesis, a foundational topic in the field of dietary oils like DHA and EPA, which are essential in modern nutrition" (Lines 239-243) based on the journals in which the papers were published?
Response: Thank you very much. We have revised and improved with new information (lines 288 to 290).
Yes, the downloaded articles indicates significant progress in crop lipid biosynthesis, with researchers actively engaged and dedicating substantial efforts to this field.
Comment #10: "Specifically, crop essential fatty acids synthesis is rooted in lipid biosynthesis and serves as a critical link between food science engineering and nutrition, contributing to a healthier food system" (Line 247). What is the evidence for the claim that it "...serves as a critical link between..."?
Response: Thank you very much for your valuable suggestion. We have rewritten and modified the information in revised manuscript (Line 298-299).
Our evidences are as alarming population growth, While in contrast fishes and marine ecosystems which are basic sources of essential PUFAs are limited, and also strict policies and regulations, example: International regulations on shark fishing bans, along with stricter water and environmental policies, have prompted a shift toward alternative sources of oils. As a result, there is increasing emphasis on developing crop-based oils to replace marine-derived sources, ensuring sustainability while meeting global demand for essential fatty acids.
Comment #11: Figure 3 does not show any connecting lines.
Response: we have modified the figure 3.
Comment #12: "Plant-based sources of essential fatty acids are now seen as the primary alternative to meet growing market demand because of the international ban on shark fishing" (Line 267). What is the basis for this claim by the authors?
Response: Thank you. We have modified it.
With respect to your suggestion, our analysis of various downloaded PDFs, the primary alternative to meet the demand for essential fatty acids is the development of sustainable, plant-based, and microbial-derived oil sources. Key findings include: (1) the U.S. implementation of shark-finning bans to promote sustainable fisheries, (2) the impact of overfishing on shark populations, leading to increased conservation efforts, (3) advancements in genetically modified crops, such as Camelina sativa, which facilitate the production of omega-3 fatty acids as a viable alternative to fish oil, (4) the emergence of Ahiflower oil as a sustainable plant-based omega-3 source, and (5) the development of microbial and genetically engineered oils as replacements for fish oil in aquaculture, supporting long-term sustainability. These studies collectively highlight the transition towards alternative sources to fulfil the demand of essential fatty acid requirements.
Comment #13: Some journal information in Table 3 may be incorrect (e.g., Plant Biotechnology).
Response: Thank you for your support; we appreciate it. The corrections have been made successfully.
Comment #14: Table 4 lacks units for some metrics.
Response: We have improved Table 4 as per suggestion.
Comment #15: "3.9. Study Limitations: One limitation of our study is that publications were extensively sourced from databases such as PubMed, Web of Science, Semantic Scholar, and Google Scholar. These databases were selected due to their broad coverage of biomedical and crop biotechnology literature. A second limitation is that the documents were collected only within the time frame from 2014 to 2024, potentially restricting the scope of earlier relevant research" (Lines 279-284). These limitations could clearly be addressed through adjustments in the search methodology. Why do the authors consider these aspects as limitations, and why did they proceed with this approach for literature retrieval and analysis?
Response: Thank you for the keen observation. We have revised our study limitation section and omit the previous lines in the revised manuscript. (Lines 331 -338).

Reviewer 3 Report
Comments and Suggestions for Authors
The contribution investigates important PUFA research through bibliometric analyses, but fails in locating influential or good works. In addition, many misunderstandings exist, revealing the lack of plant knowledge of the authors. In fact, this is not at all a review on crop biosynthesis.
Major:
1. Study plant lipids
The description of fatty acid biology is not correct. For example, microsome (line 78) is not really an organelle. The reaction of 3-hydroxyacyl-CoA dehydratase is for 3-hydroxyacyl-CoA, not ACP. PPAR and LXRs do not exist in plants.
2. Why crops?
There is no justification for the selection of 73 papers. In fact the number is very small. I do not think " oil rich crops" are fairly selected from this small set. Let alone their biosynthesis.
3. Responsible genes
The number of genes mentioned is very small and they are species-specific. Please discuss more on orthologs and their overall distribution in each crop genomes so that we can understand how oil crops have been selected in agriculture.
Author Response
Reviewer #3
The contribution investigates important PUFA research through bibliometric analyses, but fails in locating influential or good works. In addition, many misunderstandings exist, revealing the lack of plant knowledge of the authors. In fact, this is not at all a review on crop biosynthesis.
> Recommendation: Major.
Comment #1: Study plant lipids
The description of fatty acid biology is not correct. For example, microsome (line 78) is not really an organelle. The reaction of 3-hydroxyacyl-CoA dehydratase is for 3-hydroxyacyl-CoA, not ACP. PPAR and LXRs do not exist in plants.
Response: Sorry for the inconvenience. We have improved our knowledge and made corrections as per suggestion (see line 75-116)
Comment #2: Why crops?
There is no justification for the selection of 73 papers. In fact the number is very small. I do not think " oil rich crops" are fairly selected from this small set. Let alone their biosynthesis.
Response: Thank you for your valuable feedback. Since this study follows systematic and mixed review approaches, we applied both systematic and bibliometric review methods. Accordingly, 21 articles were utilized to generate original information, while 73 articles were included for bibliometric analysis. This approach was intentionally adopted to enhance the core information beyond the original research articles.We have selected 73 papers from databases and selected 21 original research article according to inclusion and exclusion criteria in time period of 2014-2024. We have focused on just oil rich crops in this time period. Also there is increasing emphasis on developing crop-based oils to replace marine-derived sources, ensuring sustainability while meeting global demand for essential fatty acids.
Comment #3: Responsible genes
The number of genes mentioned is very small, and they are species-specific. Please discuss more on orthologues and their overall distribution in each crop genome so that we can understand how oil crops have been selected in agriculture.
Response: Thank you so much for your precious suggestion. We have tried to discuss orthologues in the revised manuscript as Line (281-286). Even though the number are still less because of our data restriction.

Round 2
Reviewer 2 Report
Comments and Suggestions for Authors
First, I would like to thank the authors for addressing all the questions raised in detail. The revised manuscript has been significantly improved, but some issues remain. For example, the journal information listed (e.g., "Plant biotechnology") still contains inaccuracies, particularly regarding journal names and their corresponding impact factors (though I believe listing such details is unnecessary). Additionally, the limitations stated in Section 3.9 ("Study Limitations")—such as the restricted timeframe for literature retrieval—could have been mitigated through more systematic planning in the preliminary literature review. As a review article, analyzing only 21 papers severely limits the validity of conclusions and provides minimal reference value for readers. The authors are advised to avoid such shortcomings in future work.
Author Response
Point-by-point response letter
Respected Editor-in-Chief 16/March /2025
“International Journal of Molecular Sciences”
Thank you for your valuable comments and for giving us an opportunity to revise our review research paper “Advancements in Crop PUFAs Biosynthesis and Genetic Engineering: A systematic and mixed review system”, Manuscript ID: ijms-3469814.
The manuscript has now been revised fully as suggested by the Editor and reviewers. Response to the reviewer’s comments has been made. Appropriate changes have been inserted within the re-submitted manuscript and the changes are highlighted in red.
Reviewers' comments:
Comments and Suggestions for Authors
First, I would like to thank the authors for addressing all the questions raised in detail. The revised manuscript has been significantly improved, but some issues remain. For example, the journal information listed (e.g., "Plant biotechnology") still contains inaccuracies, particularly regarding journal names and their corresponding impact factors (though I believe listing such details is unnecessary). Additionally, the limitations stated in Section 3.9 ("Study Limitations")—such as the restricted timeframe for literature retrieval—could have been mitigated through more systematic planning in the preliminary literature review. As a review article, analyzing only 21 papers severely limits the validity of conclusions and provides minimal reference value for readers. The authors are advised to avoid such shortcomings in future work.
Response:
We acknowledge the inaccuracies in the journal names and impact factors and have made the necessary corrections to improve accuracy.
While our study aimed to focus on the most recent and relevant research, we recognize the concern regarding the limited number of papers reviewed.
We really appreciate your suggestion. Surely, we will keep these points in our mind for future work. We have modified the journal information as suggested.